# Removal of Banana Tree Fungi Using Green Tuff Rock Powder Waste Containing Zeolite

**Toyohisa Fujita [1],\* , Josiane Ponou [2], Gjergj Dodbiba [2], Ji-Whahn Anh [3], Siminig Lu [1], Mohammed F. Hamza [1] and Yuezou Wei [1],\***

[1] College of Resources, Environment and Materials, Guangxi University, Nanning 530004, China; siminglu@st.gxu.edu.cn (S.L.); m_fouda21@hotmail.com (M.F.H.)
[2] Graduate School of Engineering, The University of Tokyo, Tokyo 113-8656, Japan; ponou@sys.t.u-tokyo.ac.jp (J.P.); dodbiba@sys.t.u-tokyo.ac.jp (G.D.)
[3] Korea Institute of Geoscience and Mineral Resources (KIGAM), Daejeon 34132, Korea; ahnjw@kigam.re.kr
**\*** Correspondence: fujitatoyohisa@gxu.edu.cn (T.F.); yzwei@gxu.edu.cn (Y.W.)

**Abstract:** Hinai green tuff, which is found in Akita Prefecture, Japan, is used for the production of building materials, etc. About 60% of all stone is emitted as waste powder and therefore it is important to find ways for recycling it. In this work, the characteristics of green tuff powder have been investigated. The results of scanning electron microscope (SEM) and elemental map observations indicate that the green tuff contains $TiO_2$ on zeolite. The green tuff can therefore be used as a natural catalyst for producing hydrogen peroxide with moisture and oxygen with light. The optimum calcined temperature of the green tuff powder is about 800 °C, producing the hydroxyl radical from hydrogen peroxide decomposition without ultraviolet light (UV) and decomposition of the superoxide anion. As the application of green tuff powder, Cavendish banana trees found in the Philippines infected by a new Panama disease were treated with powder suspension in order to remove the fungus (a type of Fusarium wilt) due to the photocatalyst characteristics of powder. The suspension, prepared by using the powder was sprayed on the infected banana trees for about one month. Photograph observation indicated that the so-called 800 °C suspension spray was more effective in growing the infected banana trees.

**Keywords:** green tuff; calcination; photocatalyst; $TiO_2$; zeolite; banana tree; new Panama disease; Fusarium; SEM; radical; Philippines

## 1. Introduction

Green tuff is a sedimentary rock formed in rivers and lakes from 20 million to 15 million years ago and distributed widely along the Sea of Japan coast of the Japanese archipelago [1]. Nowadays, Hinai green tuff, a beautiful greenish color block (Towada stone) found in Akita Prefecture, Japan, is utilized in building walls and floors, etc. [2]. However, when the tuff stone is quarried and ground, about 60% of all quarried stone becomes waste powder and about 4000 t of powder per year have been discharged and wasted. It is important to utilize large amounts of produced green tuff powder. Several applications have been suggested for using the cutting powder of green tuff, for example, utilization in food processing [3], activation of microorganisms [4], as a reagents for precipitation impurities during a wastewater treatment [5], as an adsorbent for adsorption of chemical substances [6], and for the removal of formaldehyde [7]. The zeolitic adsorbent is synthesized from green tuff by hydrothermal treatment and the silver ion adsorption has been studied [8]. As the application of a natural zeolitic tuff (Nereju, Romania), a textile dye adsorption characteristic was studied [9]. The clinoptilolite-type zeolite is found in part of the green tuff belt of Northern Iran [10].



Davari et al. reported that the synthesized $ZnO/Fe_2O_3$ and $TiO_2/Fe_2O_3$ on zeolite could decompose organic substance [11]. ZnO nanoparticles are effective on fungi like Fusarium as antimicrobial agents [12]. Complexation of $TiO_2$ particles with rutin shifts the photogeneration of hydroperoxyl (HOO) and hydroxyl (HO) radicals toward visible light (lambda > 400 nm) [13]. As the green tuff contains small amounts of similar components, the elemental distribution is measured and investigated especially for $TiO_2$ existence on zeolite. If the natural photocatalyst is possible, the synthetic photocatalyst production cost can be reduced. Photocatalysis describes the excitation of titanium dioxide nanoparticles (a wide-band gap semiconductor) by UV light to produce reactive oxygen species (ROS) that can destroy many organic molecules including fungi by the addition of inorganic salt potassium iodide [14]. Many kinds of photocatalytic productions of hydrogen peroxide on a semiconductor, such as $TiO_2$, have been reported using water and oxygen with UV, however, hydrogen peroxide can be produced by using visible light [15,16]. The produced hydrogen peroxide can be decomposed by higher alkaline [17], UV light irradiation, and the Fenton reaction with the $Fe^{2+}$ ion. For the application of this phenomena, the removal of banana tree fungi is investigated.

Banana (*Musa* spp.) is a staple food for more than 400 million people, and over 40% of world production and virtually all the export trade is based on the Cavendish banana. However, the Cavendish banana is under threat from a virulent fungus, *Fusarium oxysporum* f. sp. *cubense* (FOC), Tropical race 4 (TR4), for which no acceptable resistant replacement has been identified [18]. FOC infection causes the oxidative stress to the banana and the development of stacked antimicrobial genes in the banana also is studied [19]. The photograph in Figure 1 shows the early symptoms such as wilting and leaf-yellowing of the banana tree in this experimental field in the Philippines. An typical example of an infected banana tree is shown in the reference [18], which indicates that the disease caused the yellowing and dropping of the leaves, known as the pseudostem cracking. The typical example shows more brown leaf structure and plant collapse. In the experiment, the suspensions of green tuff powder and calcined green tuff powder were sprayed on the infected banana trees periodically and the situation of the banana trees was observed after about one month. As the green tuff does not dissolve in water, the green tuff powder and water suspension are environmentally friendly materials.

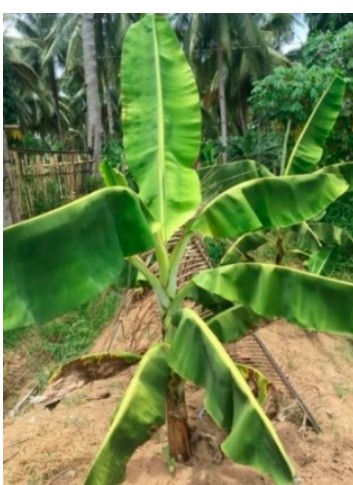

**Figure 1.** Photograph of early symptoms such as wilting and leaf-yellowing of a banana tree in this experimental field in the Philippines.

## 2. Results and Discussion

### 2.1. Calcined Green Tuff Powder

The green tuff powder was calcined at temperatures ranging from 200 °C to 1200 °C for 30 min, and the calcined powder was then kept dry. XRD patterns of each calcined powder are shown in Figure 2, which indicates mainly quartz and albite peaks. Chlorite peaks exist in the green tuff calcined

at 500 °C, however, they disappear at temperatures higher than 600 °C. Green color chlorite disappears by calcination at higher than 600 °C and $Fe^{2+}$ in chlorite has oxidized to $Fe^{3+}$. Albite is stable at all calcined temperatures. Stilbite and laumontite exist at temperatures lower than 700 °C and they disappear at temperatures higher than 800 °C. Both zeolite of stilbite and laumontite contain Ca and as they decompose, small peaks of CaO appear between 600 °C and 800 °C, while it was disappeared at temperature above 900 °C.

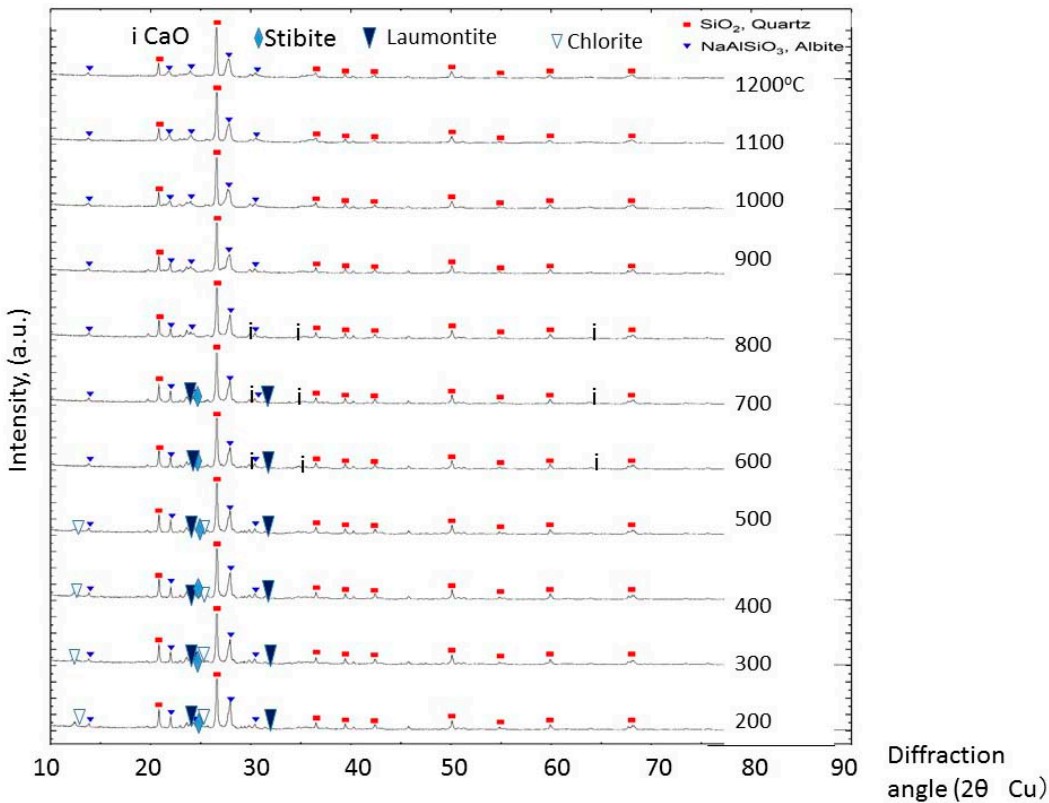

**Figure 2.** XRD of green tuff after calcination at each temperature for 30 min.

Next, the green tuff powder and the shape of the calcined powder were investigated by SEM, and results are shown in Figure 3, which shows that: 1. the fine ground green tuff powder has coagulated as calcination temperature increases; and, 2. the particle size of 1100 °C calcined powder becomes larger. The compared specific surface areas and pore diameters of green tuff powder and 800 °C calcined powder are listed in Table 1, while Figure 4 shows the mesoporous size distribution of green tuff powder and 800 °C calcined powder using the Saito–Foley (SF) method. The specific surface area 16.79 $m^2$/g of green tuff powder decreased to 6.25 $m^2$/g by calcination at 800 °C, however, the average diameter of the micropore was the same, while the mesopore size and surface area of calcined powder decreased comparing to the as-received green tuff powder. The fungus Fusarium cannot enter the pores of porous green tuff for the larger size of fungus, comparing with tuff pore size shown in Table 1.

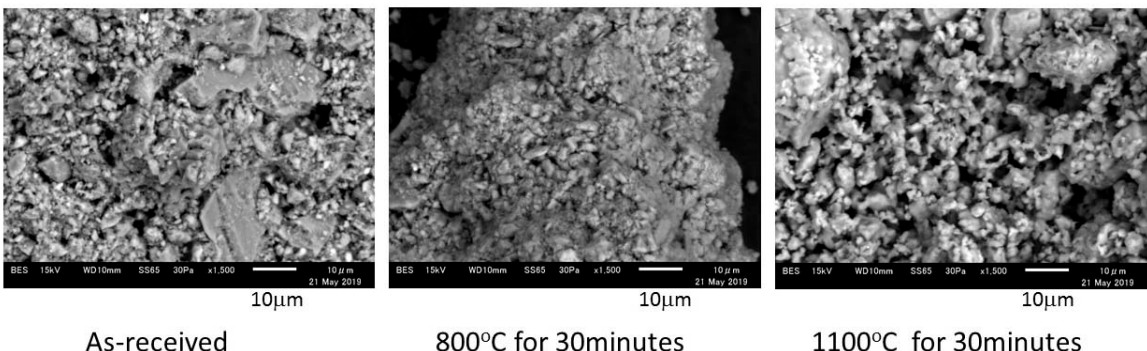

As-received          800°C for 30minutes          1100°C for 30minutes

**Figure 3.** Photographs of as-received green tuff powder and calcined powder at 800 °C and 1100 °C for 30 min.

**Table 1.** Specific surface area and pore diameter of green tuff powder and 800 °C calcined powder.

|  | Type | As-Received | 800 °C Calcined |
|---|---|---|---|
| Surface area, (m²/g) | BET | 16.79 | 6.25 |
|  | Mesopores | 11.65 | 4.68 |
| Pore Diameter, (nm) | Average | 10.00 | 21.97 |
|  | Mesopores | 3.38 | 3.06 |
|  | Micropores | 0.45 | 0.45 |

Micropores: 0–2 nm; Mesopores: 2–100 n; Micropores area-diameter have been calculated using SF method; Mesopores are-diameter have been calculated using BJH method.

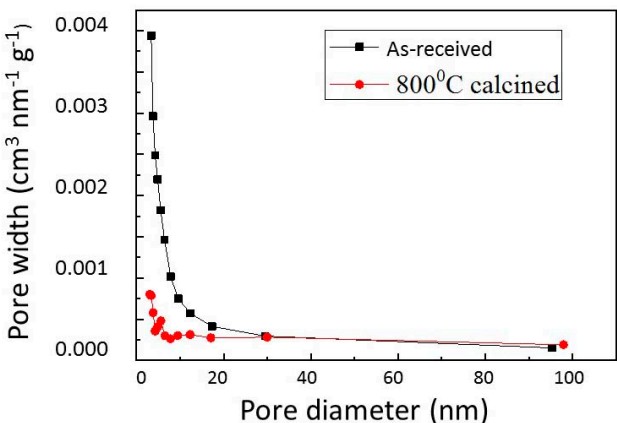

**Figure 4.** Mesopore size distribution of green tuff powder and 800 °C calcined powder using Saito and Foley (SF) method.

Figure 5 shows the Ca and Ti elemental maps of the powder calcined at 800 °C compared with the original green tuff particles. The corresponding places of Ca and Ti are shown in the circle marks. Considering the mineral composition shown in Table 1, the Ca map shows mainly zeolite in Table 1. A concentrated Ti map corresponds to the Ca map and this phenomenon is observed in all different calcined green tuff samples [7]. The fine grains of $TiO_2$ on zeolite may be a photocatalyst in any green tuff.

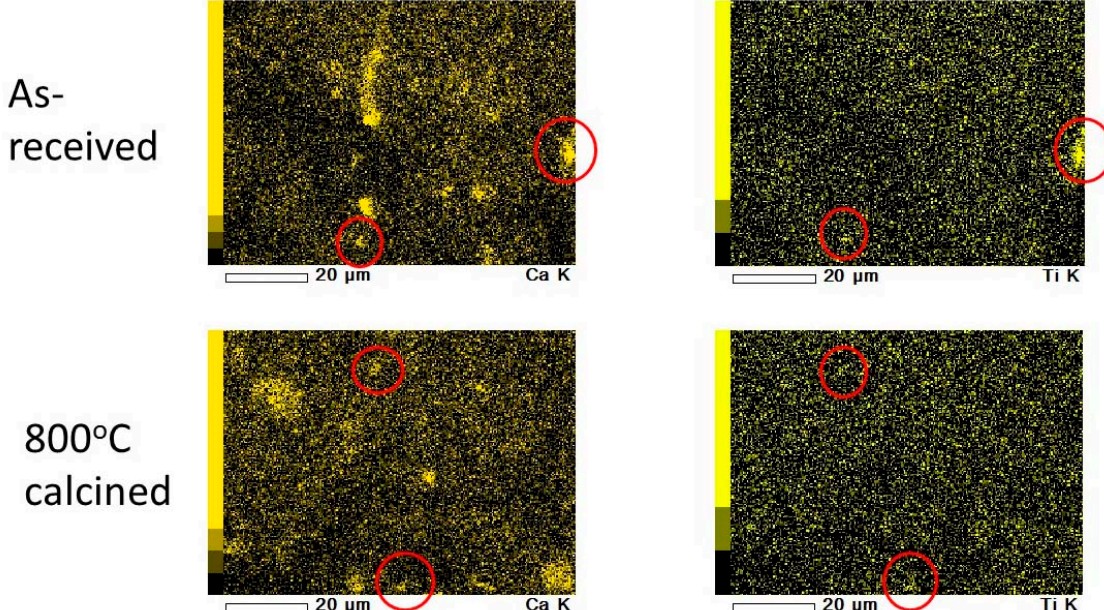

**Figure 5.** EDS elemental maps of Ca and Ti for as-received green tuff (top two photos) and 800 °C calcined green tuff powder (bottom two photos). The Ca and Ti corresponding places are shown in the red circle marks.

## 2.2. Radical Measurement by Adding Green Tuff and Calcined Powder

A general scheme for the production of reactive oxygen species where oxygen is acting as the electron acceptor and water or hydroxyl ions act as the electron donor is shown in Figure 6 [20]. Here, $TiO_2$ is the semiconductor. The valence band hole may have an electrochemical reduction potential positive enough to oxidize water for hydroxyl radical production and the conduction band should be negative enough to reduce molecular oxygen for production the superoxide anion radical. The oxygen and water (as a moisture) on the $TiO_2$ in tuff powder may cause the above reaction and a mixture of reactive oxygen species (ROS) has a possibility of killing the fungi on the banana leaves. The mechanism of photocatalytic inactivation of microorganisms is reviewed by Byrne et al. [20]. If $TiO_2$ on zeolite in green tuff and calcined ones is a photocatalyst, the hydrogen peroxide can be produced in the following equation as positive hole $h^+$ [15,21].

$$H_2O + 2\,h^+ \rightarrow 1/2O_2 + 2H^+ \tag{1}$$

$$O_2 + 2H^+ + 2e^- \rightarrow H_2O_2 \tag{2}$$

By irradiating UV light, $H_2O_2$ produces the hydroxyl radical (·OH) in the following equation [22]. Here, $h$ is Planck's constant and $v$ is light frequency

$$H_2O_2 + hv = 2\,\text{·OH}. \tag{3}$$

Instead of the production radical, in this experiment the reaction of the already prepared oxygen species ($H_2O_2$, OH, $O_2^-$) in water by adding green tuff and calcined powders are investigated. The effect of pH on the calcined temperature of green tuff by dispersing 1 wt% of powder in water is shown in Figure 7. The green tuff powder calcined at around 800 °C shows higher pH compared with the original green tuff one, while the higher than 1100 °C calcined tuff powder suspension shows lower pH. As shown in Figure 2 the CaO (small peaks) is observed from 600 °C to 800 °C and that dissolved is responsible for increasing the pH. The pH increase by calcination is similar to the calcined dolomite [23]. The hydrogen peroxide is unstable at higher pH [17], it means that the alkaline solution can decompose it, and it is noteworthy that the 800 °C calcined green tuff has a large decomposition ability.

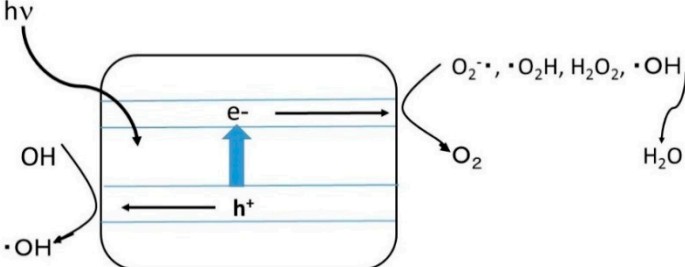

**Figure 6.** Schematic of photocatalytic mechanism on a titanium dioxide particle leading to the production of reactive oxygen species [20].

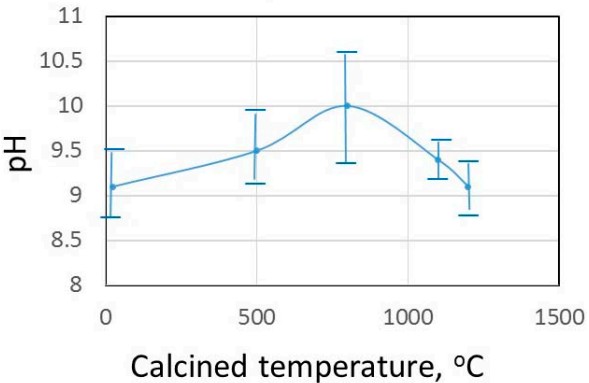

**Figure 7.** Effect of pH on the calcined temperature of green tuff by dispersing 1wt% of powder in water.

ESR spectra with or without UV illuminated for hydrogen peroxide solution (0.1 w/v%) for as-received green tuff as well as 800 °C calcined green tuff is shown in Figure 8. Only $H_2O_2$ solution does not produce a hydroxyl radical, however, the 800 °C calcined tuff powder addition produces the hydroxyl radical without UV irradiation. Under UV, both solutions showed the hydroxyl radical peaks largely, while by the addition of calcined powder, the hydroxyl radical peak is smaller than only the $H_2O_2$ solution. $H_2O_2$ decomposition to hydroxyl radical has already started with the addition of powder.

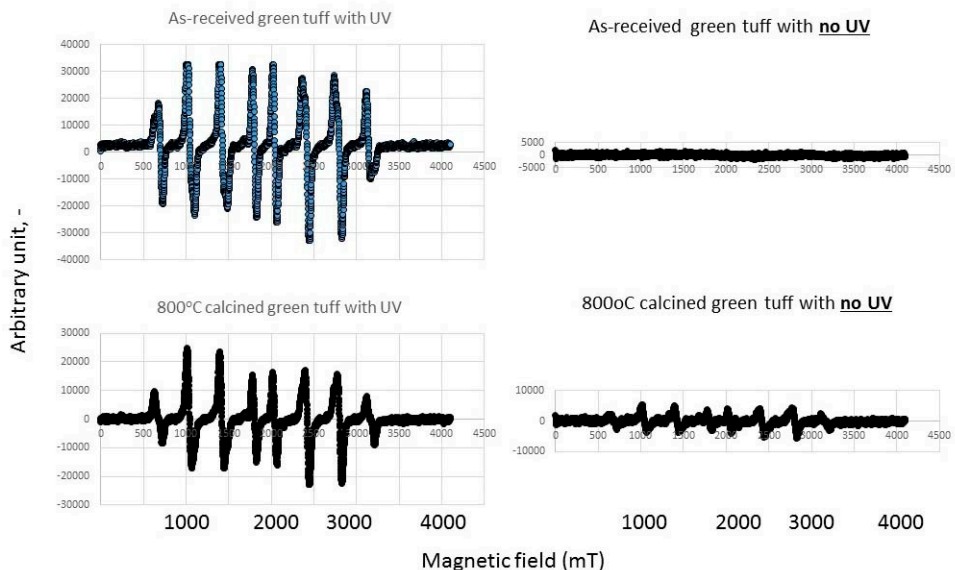

**Figure 8.** ESR spectra of G-CYPMPO-trapped adduct in UV or no UV illuminated hydrogen peroxide solution (0.1 w/v%) for as-received green tuff (above) and for 800 °C calcined green tuff (below).

ESR spectra in superoxide solution, 800 °C calcined green tuff, and no additive of powder as background are shown in Figure 9. The additives of calcined powder decrease the superoxide anion compared with no additive powder background with no UV. The additives of 800 °C calcined green tuff powder can react faster with the produced $H_2O_2$ and superoxide anion. The produced hydroxyl radical on the 800 °C calcined tuff might kill the fungus on the banana trees by photocatalysis.

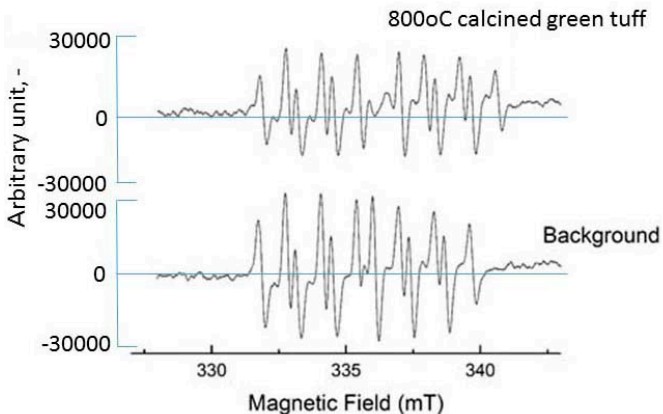

**Figure 9.** ESR spectra of G-CYPMPO-trapped adduct in superoxide solution (above) and for 800 °C calcined green tuff (above) and no additive of powder as background (below).

### 2.3. Result of Infected Banana Trees after about One Month

The infected banana tree after being sprayed 10 times for about one month is shown in Figure 10. The brown color suspension spray containing 800 °C calcined tuff powder showed better leaf growth than the green color suspension spray containing as-received original green tuff spray. The fungus *Fusarium oxysporum* f. sp. *cubense* (Foc) on the petiole of the leaf blade, throat of the plant, and leaf sheaths in Figure 12 might have been killed by photocatalysis of 800 °C calcined brown tuff powder.

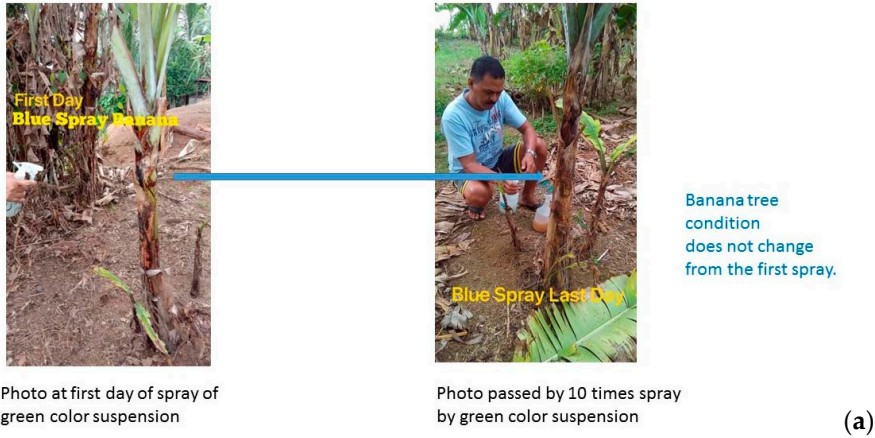

Photo at first day of spray of green color suspension

Photo passed by 10 times spray by green color suspension

Banana tree condition does not change from the first spray.

**(a)**

**Figure 10.** *Cont.*

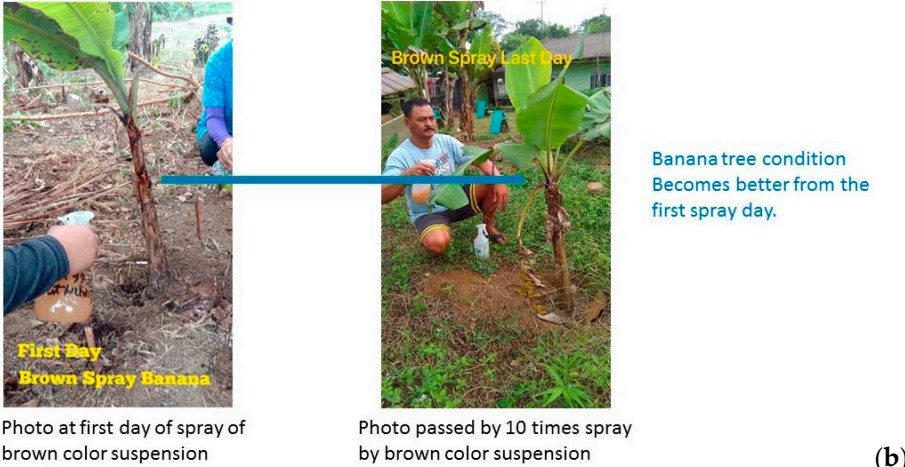

Banana tree condition Becomes better from the first spray day.

Photo at first day of spray of brown color suspension

Photo passed by 10 times spray by brown color suspension

(**b**)

**Figure 10.** The infected banana tree after 10 spray times for about one month. (**a**) Original green tuff suspension, (**b**) 800 °C calcined tuff suspension.

## 3. Materials and Experiment

### 3.1. Green Tuff Powder Characteristics

The green tuff powder is collected from the Towada stone cutting place in the quarry to produce the architectural materials for walls and flat floor tiles in Odate city in Akita Prefecture, Japan. The composition of the green tuff cutting waste powder less than −100 mesh in size is listed in Table 2. The composition was analyzed by X-ray fluorescence spectroscope (XRF). It was found that the iron oxide is mainly ferrous. The main composition minerals are albite ($NaAlSi_3O_8$) 35%, quartz 25%, chlorite ($(Mg,Fe,Al)_6(Si,Al)_4O_{10}(OH)_8$) 7% and zeolite (stilbite 3% and laumontite 2%), identified by X-ray diffraction (XRD) and the average density is 2.1 g/cm [4]. The zero point of charge ($pH_{pzc}$) is about pH 2.5 [5]. Next, the green tuff powder was calcined at 200 °C to 1200 °C for 30 min and the calcined powder was kept dry. The shape and elemental distribution of the green tuff powder and the calcined powder were investigated by SEM and Energy Dispersive X-ray Spectroscopy (EDS).

**Table 2.** Chemical and mineral composition of green tuff powder [4].

| Composition | Content ,w/v % |
|---|---|
| $Na_2O$ | 0.30 |
| $MgO$ | 0.73 |
| $Al_2O_3$ | 9.16 |
| $SiO_2$ | 61.69 |
| $P_2O_5$ | 0.21 |
| $SO_3$ | 0.17 |
| $Cl$ | 0.04 |
| $K_2O$ | 6.19 |
| $CaO$ | 4.07 |
| $TiO_2$ | 0.71 |
| $MnO$ | 0.49 |
| $FeO$ or $Fe_2O_3$ | 14.42~16.03 |
| $Co_2O_3$ | 0.01 |
| $ZnO$ | 0.07 |
| $Rb_2O$ | 0.03 |
| $SrO$ | 0.08 |
| Total | 98.37~99.98 |

7% 28% 25% 5% 35%

2% 3%

■ Albite ■ Quartz ■ Chlorite ■ Others ■ Zeolite

■ Stilebite ■ Laumontite

Zeolite
Stilbite ($CaAl_2Si_7O_{18}7H_2O$)
Laumontite ($Ca_4Al_8Si_{16}O_{48}18H_2O$)

### 3.2. Radical Measurement by Electron Spin Resonance (ESR)

The effects of a green tuff powder addition into hydroxyl radical and superoxide radical solutions were investigated using ESR. A JEOL JES-TE25X ESR spectrometer was used and typical ESR measurement conditions were as follows: microwave power, 4 mW; microwave frequency, 9.2 GHz; magnetic field, 328.0 mT; field sweep width, ±7.5 mT; field modulation, 0.16 mT; sweep time, 1 min; 0.003663 mT/Point, 4096 points in total. The measurements were performed at room temperature. To investigate the effect of the radicals, 2% w/v green tuff powder was mixed in two kinds of solution. One contained 0.1% w/t hydrogen peroxide aqueous solution to produce the hydroxyl radical with ultraviolet light (UV) as well as without UV. The other solution was a hypoxanthine/xanthine oxidase (HX/XO) system solution to produce the superoxide radical (mixture of 1.8 vol% of 10.97 units/mL XO and 10 vol% of 20 mM HX). In this study, a novel radical trapper, G-CYPMPO [24], was used for trapping the free radicals.

### 3.3. Fusarium Wilt and Experimental Method in a Banana Plants in Luzon Island, Philippines

Banana production is seriously threatened by Fusarium wilt, a disease caused by the soil-borne fungus *Fusarium oxysporum* f. sp. *cubense* (Foc). Foc TR4 has been restricted to the East and parts of Southeast Asia for more than 20 years, but since 2010 the disease has spread westward into five additional countries in Southeast and South Asia and at the transcontinental level into the Middle East and Africa (Mozambique) [25]. Nowadays, the most common banana subgroup is the Cavendish, which makes up most of the global market. There are large banana plantations in Mindanao Island, Philippines, as well Fusarium wilt problems.

Fusarium wilt is considered as one of the most important and destructive diseases in banana crops worldwide. Also of concern is the spread of another fungal disease, black Sigatoka, whose spores travel through the air, causing infecting plants and reducing fruit yields. Climate change also assists in the spread of this fungus. The uptick in weather conditions favorable to black Sigatoka has boosted the risk of infection by almost 50% since 1960 in some parts of the world [26]. The spread of Foc TR4 is of great concern due to the limited knowledge about key aspects of the disease epidemiology and the lack of effective management models, including resistant varieties and soil management approaches. The infection process, plant–pathogen interaction, and disease developments have been reported by Dita et al. that shows the lifecycle of *Fusarium oxysporum* f. sp. *clubense* (Foc) in bananas [25]. An actinobacteria strain (named SCA3-4) was screened against *Fusarium oxysporum* f. sp. *cubense*, Tropical race 4 (Foc TR4, ATCC 76255) [27] and a photograph of the mycelium of the actinobacteria strain is shown in Figure 11. The spherical spore size is about 1 μm and strain width ranged from 0.2 to 0.5 μm. Fusarium wilt will not grow in a pore size smaller than 100 nm.

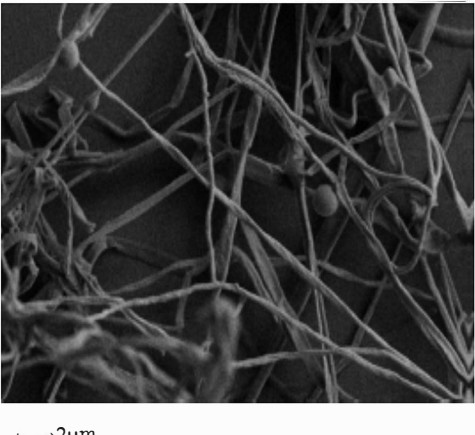

←——→2μm

**Figure 11.** Mycelium of an actinobacteria strain screened against *Fusarim oxysporum* f. sp. *cubense*, Tropical race 4 (Foc TR4, ATCC 76255) [27].

The banana plants were dissected in two places as shown in Figure 12; first directly under the throat (Figure 12, zone D), and second, at the soil level (Figure 12, zone F) approximately 2 cm above the rhizome [28]. Within the non-senescing leaf sheaths, the migration of Foc was confined to the xylem vessels and hyphal growth was apparent on the outer surface of senescing leaf sheaths. In this experiment, the green tuff suspension and 800 °C calcined tuff suspension were sprayed in the throat of the plant (D) and leaf sheaths (E) as in Figure 13 for killing the fungus. The suspensions were prepared as follows: 1 wt% of −100-mesh size green tuff powder and calcined powder at 800 °C for 30 min were mixed well with water and kept in the vessel. Before the suspension was sprayed on the banana trees, it was well-agitated and then sprayed on the infected banana trees (500 cc per one time). The spraying times are shown in Figure 13. The experimental place was Barangay Hoyo, Silang, Cavite, Philippines.

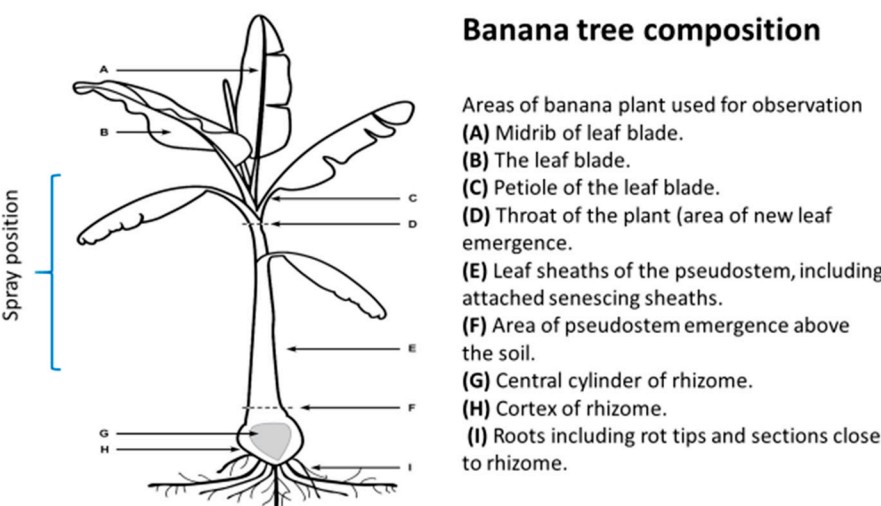

**Banana tree composition**

Areas of banana plant used for observation
**(A)** Midrib of leaf blade.
**(B)** The leaf blade.
**(C)** Petiole of the leaf blade.
**(D)** Throat of the plant (area of new leaf emergence.
**(E)** Leaf sheaths of the pseudostem, including attached senescing sheaths.
**(F)** Area of pseudostem emergence above the soil.
**(G)** Central cylinder of rhizome.
**(H)** Cortex of rhizome.
**(I)** Roots including rot tips and sections close to rhizome.

**Figure 12.** Banana tree composition [28] and the spray areas in this experiment.

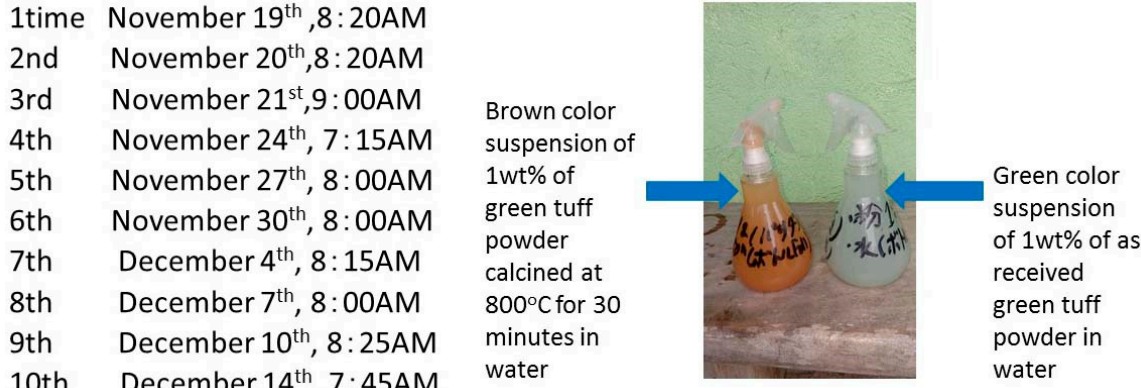

| | |
|---|---|
| 1time | November 19th, 8:20AM |
| 2nd | November 20th, 8:20AM |
| 3rd | November 21st, 9:00AM |
| 4th | November 24th, 7:15AM |
| 5th | November 27th, 8:00AM |
| 6th | November 30th, 8:00AM |
| 7th | December 4th, 8:15AM |
| 8th | December 7th, 8:00AM |
| 9th | December 10th, 8:25AM |
| 10th | December 14th, 7:45AM |

Brown color suspension of 1wt% of green tuff powder calcined at 800°C for 30 minutes in water

Green color suspension of 1wt% of as received green tuff powder in water

**Figure 13.** Spraying times and photographs of two kinds of suspension. (Green color suspension dispersing as-received green tuff powder in water and brown color suspension dispersing 800 °C 30 min calcined green tuff powder).

## 4. Conclusions

The wasted green tuff powder produced by cutting Towada stone was analyzed and utilized to kill the fungus on the banana trees on a Philippine farm. The fine $TiO_2$ that exists on zeolite in green tuff powder was treated with the calcined powder, which is considered to be a natural photocatalyst. The 800 °C calcined green tuff powder decomposed hydrogen peroxide and superoxide anions. The brown color suspension spray containing 800 °C calcined tuff powder showed better leaf growth than the green color suspension spray containing as-received green tuff spray after 10 spray times for about one

month. The fungus *Fusarium oxysporum* f. sp. *cubense* (Foc) on the petiole of the leaf blade, throat of the plant, and leaf sheaths might have been killed by photocatalysis of 800 °C calcined brown tuff powder.

**Author Contributions:** Conceptualization, T.F.; investigation, J.P., S.L.; writing—original draft preparation, T.F.; review and editing, G.D., M.F.H.; supervision, Y.W., J.-W.A.

**Funding:** This research received no external funding

**Acknowledgments:** We appreciate Kazumi Otabe, Econergy Co. Ltd., and Romila Rivera, ALIMOR International Trading, for the experiment in the Philippines. Also, we have deep appreciation for graduate student Lanyin Zhang in the University of Tokyo, Hiromi Kurokawa and Hirofumi Matsui of the University of Tsukuba. The green tuff was supplied by Shigeki Yamamoto and Hiroshi Kawaguchi in Towada Green-tuff Agro Science Co., Ltd.

**Conflicts of Interest:** The authors declare no conflict of interest.

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
