# Peer review of "Removal of Banana Tree Fungi Using Green Tuff Rock Powder Waste Containing Zeolite"

_catalysts, doi:10.3390/catal9121049_

Round 1

Reviewer 1 Report

In this reviewer opinion the research topic of paper under review seems to be within the scope of the journal. However, the major revision is recommended prior to its publication. Here, some issues that must be addressed:

The purpose and novelty of the paper should be highlighted. The authors should thoroughly search and cite all the newest publications on this research topic.

Conclusion part needs to be revised as the experimental results from Results and discussion section should not be repeated in this section.

The lacking key words have to be completed.

All typing mistakes should be corrected.

In some sentences English needs to be revised to reach the language standards that are required by the journal.

Author Response

Thank you very much for reviewing. The paper was corrected as follows,

・The purpose and novelty of the paper should be highlighted. The authors should thoroughly search and cite all the newest publications on this research topic.

In 40-43,46,53 line: The recent publication references [8], [9], [10],[12],[13] are added in the introduction.

・Conclusion part needs to be revised as the experimental results from Results and discussion section should not be repeated in this section.

Conclusion parts was revised simply by eliminating repeated parts as follows.

The wasted green tuff powder produced by cutting Towada stone was analyzed and utilized to kill the fungus on banana trees in the farm of Philippine. The fine TiO2 exists on zeolite in green tuff powder was considered with the calcined powder as a natural photocatalyst. The 800oC calcined green tuff powder decomposed hydrogen peroxide and superoxide anions. The brown colour suspension spray containing 800 oC calcined tuff powder showed the better leaf growth than the green colour suspension spray containing as-received green tuff spray after 10 times spray for about one month. The fungus Fusarium oxysporum f. sp. cubense (Foc) on the petiole of the leaf blade, throat of the plant and leaf sheaths might be killed by photocatalysis of 800oC calcined brown tuff powder.

・The lacking key words have to be completed.

Key words of SEM, radical, Philippine are added.

・All typing mistakes should be corrected.

In some sentences English needs to be revised to reach the language standards that are required by the journal.

English words and sentences were corrected.

Reviewer 2 Report

This manuscript provides useful application about Hinai green tuff found in Akita Prefecture, Japan as natural catalyst. The green tuff powder was characterized using XRD, SEM-EDS and BET surface area, pore size analysis. The green tuff was utilized to remove fungus of banana trees due to its photocatalyst characteristics. The methodological approach seems solid and appropriate. The paper was well organized, and the results are important for environmental science and ecological studies. There is no problem regarding English. Therefore, I feel this paper should be acceptable after minor revision in view of the following specific comments.

(1) It would be better if the results are compared with other previous studies such as other catalysts. What is the most excellent merit using green tuff rock?

(2) The lettering or numerical values in Figure 5, 6, 10, 11 is not clear (or print is light), and difficult to read. Please revise.

Author Response

Thank you for reviewing. The paper was corrected as follows,

・It would be better if the results are compared with other previous studies such as other catalysts. What is the most excellent merit using green tuff rock?

From the literature survey, the next sentence is added.

Line45,46

ZnO nanoparticle is effective to the fungi like fusarium as antimicrobial agents [12]

・The lettering or numerical values in Figure 5, 6, 10, 11 is not clear (or print is light), and difficult to read. Please revise.

The figure 5 , 7 and 11 were changed. In Figure 11, the center letter is written at the bottom.

Reviewer 3 Report

                The manuscript presents a new and interesting application of natural zeolites (present in green tuff rock powder waste) for removal of banana trees fungi. It shows how wide may be application of zeolites (which are known as adsorbents, ion exchangers and catalysts in refinery industry) also in various domains of the agriculture.

                The authors presented well planned, well realized and well described results of the research which gave (as the result) positive result in medication of banana tree infected by fungi. The authors studied the material which was calcined at various temperatures. The 800 o suspension spray containing 800oC calcined material was more effective in growing the infected banana trees.

                Generally, I propose to accept the manuscript “as is”. However, as I have no experience in the ESR studies, which were done in the study, I propose to consult (if the editors find it necessary) my friend who is expert in ESR of zeolitic systems: Dr P. Pietrzyk (pietrzyk@chemia.uj.edu.pl)

                The manuscript presents a new and interesting application of natural zeolites (present in green tuff rock powder waste) for removal of banana trees fungi. It shows how wide may be application of zeolites (which are known as adsorbents, ion exchangers and catalysts in refinery industry) also in various domains of the agriculture.

                The authors presented well planned, well realized and well described results of the research which gave (as the result) positive result in medication of banana tree infected by fungi. The authors studied the material which was calcined at various temperatures. The 800 o suspension spray containing 800oC calcined material was more effective in growing the infected banana trees.

                Generally, I propose to accept the manuscript “as is”. 

Author Response

Thank you for reviewing.

The paper by Dr. Pietrzyk, P is added as reference [13]

Labuz, P ; Grybos, J  ; Pietrzyk, P  ; Sobanska, K  ; Macyk, W  ; Sojka, Z ,2019,  Photogeneration of reactive oxygen species over ultrafine TiO2 particles functionalized with rutin-ligand induced sensitization and crystallization effects, RESEARCH ON CHEMICAL INTERMEDIATES, 45, 5781-5800

Reviewer 4 Report

The manuscript from the title: "Removal of banana trees fungi using green tuff rock powder waste", is a very interesting paper. The paper is very easy to read, the experimental procedure is good, and the obtained photocatalytic powder from wasted green tuff is well characterized and the applied. In particular, application of wasted green tuff powder produced by cutting Towada stone for to kill the fungus on banana trees is very interesting idea. Reference list is up to date and more than adequate.

However, it is interesting to investigate whether the applied powder flushes during the rainfall, for example, and are heavy metals such as Mn, Fe, Co and Zn flushes during the rainfall in the soil?

Could you please clarify this point?

I recommend acceptance for publication after minor modifications.

Author Response

Thank you for reviewing.

However, it is interesting to investigate whether the applied powder flushes during the rainfall, for example, and are heavy metals such as Mn, Fe, Co and Zn flushes during the rainfall in the soil?

Could you please clarify this point?

Mn, Fe, Co and Zn shown in Table 1 does not dissolve from tuff by water of rainfall.

A lot of Fe includes in the soil and no environmental standard in the soil for Fe. Also there is no problem to use tuff as the heavy metal containing environmental standard such as Cu in the soil is less than 125mg/kg.

In Line 69 and 70, the following sentence is added.

As the green tuff does not dissolved in water, the green tuff powder and water suspension is environmental friendly materials.

Round 2

Reviewer 1 Report

In my opinion the manuscript has been revised following the previously-mentioned reviewers’ comments and suggestions and in its current version is suitable for publication